# Preparation of a Chitosan/Coal Gasification Slag Composite Membrane and Its Adsorption and Removal of Cr (VI) and RhB in Water

**DOI:** 10.3390/molecules27217173

**Published:** 2022-10-23

**Authors:** Deqiang Peng, Shuyun Zhang, Kai Wang, Tingting Dong, Min Zhang, Guohui Dong

**Affiliations:** 1College of Environmental Science and Engineering, Shaanxi University of Science and Technology, Xi’an 710021, China; 2Shaanxi Institute of Geology and Mineral Resources Experiment Co., Ltd., Xi’an 710054, China

**Keywords:** coal gasification slag, membrane material, adsorb, heavy metal, dyestuff

## Abstract

At present, there are many kinds of pollutants, including dyes and heavy metal ions, in wastewater. It is very important to develop adsorbents that can simultaneously remove heavy metal ions and dyes. In this study, a renewable composite membrane material was synthesized using chitosan and treated coal gasification slag. The Cr (VI) maximum adsorption capacity of the composite membrane was 50.0 mg/L, which was 4.3~8.8% higher than that of the chitosan membrane. For the adsorption of RhB, the removal rate of the chitosan membrane was only approximately 5.0%, but this value could be improved to 95.3% by introducing coal gasification slag. The specific surface area of the chitosan membrane could also be increased 16.2 times by the introduction of coal gasification slag. This is because coal gasification slag could open the nanopores of the chitosan membrane (from 80 μm to 110 μm). Based on the adsorption kinetics and adsorption mechanism analysis, it was found that the adsorption of Cr (VI) occurred mainly through the formation of coordination bonds with the amino groups on the molecular chains of chitosan. Meanwhile, RhB adsorption occurred through the formation of hydrogen bonds with the surface of coal gasification slag. Additionally, coal gasification slag can improve the mechanical properties of the chitosan membrane by 2.2 times, which may facilitate the practical application of the composite membrane. This study provides new insight into the adsorbent design and the resource utilization of coal gasification slag.

## 1. Introduction

In recent years, the printing, dyeing, and textile industries have developed rapidly, producing a large amount of wastewater containing dyes and heavy metal ions along with high productivity [1,2,3], but the traditional biochemical treatment technology has limited ability to decolorize dyes and remove heavy metals. The use of a proper adsorbent can not only remove the color of dyed wastewater but also remove heavy metals, which have a very important role in the whole treatment process [4]. With continuous research on adsorption treatment technology, a variety of adsorbents have been discovered and introduced, playing an increasingly significant role in the effective treatment of printing and dyeing wastewater [5]. The commonly used adsorbents include activated carbon [6], metal oxides [7], resins [8], inorganic substances [9], and natural polymeric materials [10]. However, these adsorbent materials face the problems of high wastewater treatment costs, single response to pollutants, inability to cope with complex pollutants, and limited ability to remove metal ions.

Based on this, we attempted to select a solid waste coal gasification slag with small particle size, well-developed porosity, and strong hydrophilicity as a dye adsorbent and load it on a polymeric organic membrane with a strong response to heavy metals to realize solid waste resourceification while effectively treating printing and dyeing wastewater containing heavy metal ions. Coal gasification slag is a solid waste generated in the process of coal gasification. From a component point of view, it is rich in SiO_2_, Al_2_O_3_, Fe_2_O_3_, CaO, and other inorganic minerals and residual carbon, which is the basis of resource utilization [11]. From the perspective of physical properties, the residual carbon in the gasification residue has sufficient adsorption space due to its high content, small particle size, large specific surface area, and rich pore structure [12]. Therefore, coal gasification slag was selected as the powder adsorption material in this study. Yuan et al. [13] prepared a ZSM-5 molecular sieve from coal gasification slag and used it for the adsorption of cationic dyes and found that its removal rate of methylene blue could reach 82.07%. Dong et al. [14] directly transformed industrial coal gasification slag with high heavy metal content into an excellent adsorbent for the adsorption of malachite green wastewater, and the theoretical maximum adsorption capacity reached 1787 mg/g. Ma et al. [15] used acid leaching and alkali solubilization methods to prepare carbon/zeolite composites (C/ZC) from coal gasification slag by an inducer, and the maximum adsorption capacities of the composites for NH_4_^+^ and PO_4_^3−^ were 7.44 mg/g and 6.94 mg/g, respectively, after modification by iron sulfate. Chitosan (Cs), as a product of the deacetylation of chitin [16,17], is a biodegradable natural polymer containing amino groups [18,19]. Zia et al. [20] prepared porous poly (L-lactic acid) (P-PLLA) nanofiber membranes by grafting chitosan with polydopamine (PDA) as an intermediate layer and used this material for Cu^2+^ adsorption. Jiang et al. [21] synthesized glucose/chitosan by ultrasonic-assisted free radical polymerization and adsorbed Cu (II), Co (II), and other metal ions. Therefore, we selected chitosan as the carrier of gasification residue to prepare a gasification residue/chitosan composite membrane with a special reticulation and reusable structure, and use this novel composite as a candidate to solve the problems of difficult recovery of powder adsorbent, single response of pollutants, poor adsorption of heavy metals, and high cost of wastewater treatment, and also to achieve the efficient removal of dyes and heavy metals.

In this study, the coal gasification slag was first treated at high temperatures and subsequently loaded on chitosan film to develop a chitosan/coal gasification slag composite membrane. The composite membrane was characterized by FT-IR, SEM, swelling rate, mechanical properties, specific surface area, etc. The adsorption effect of the composite membrane on Cr (VI) and RhB was investigated using Rhodamine B (RhB) and Cr (VI) as the models of dye and heavy metal in printing and dyeing wastewater. The adsorption process of the composite membrane on Cr (VI) and RhB was analyzed by kinetic simulation, and the adsorption mechanism of the composite membrane on Cr (VI) and RhB adsorption was also provided.

## 2. Experimental Section

### 2.1. Experimental Materials

The experimental materials included chitosan (deacetylated ≥ 90%, Shanghai Lanji Bioreagent Co., Ltd., Shanghai, China), glacial acetic acid (AR, Tianjin Fuyu Fine Chemical Co., Ltd., Tianjin, China), glutaraldehyde aqueous solution (25 wt%; AR, Tianjin Comio Chemical Reagent Co., Ltd., Tianjin, China), and coal gasification slag (Shaanxi Geological and Mineral Research Institute Co., Ltd., Xian, China). K_2_Cr_2_O_7_ (AR, Comio, Tianjin, China), CuSO_4_ (AR, Tianjin Tianli Chemical, Tianjin, China), Pb(NO_3_)_2_ (AR, Tianjin Tianli Chemical, Tianjin, China), and CdSO4 (AR, Tianjin Tianli Chemical, Tianjin, China) were also used in this study.

### 2.2. Preparation of Samples

Preparation of coal slag-based activated carbon: The industrial gasifier slag collected from the factory was first dried at 60 °C for 24 h, and then the dried coal slag was crushed (multi-functional grinder, HebiXinyun Equipment Co., Ltd., XY-100, Hebi, China), screened (200 mesh), and calcinated at a certain temperature in a muffle furnace (box muffle furnace, Hefei Kejing Material Technology Co., Ltd., KSL-1200X, Hefei, China) to form the small size slag. Finally, the sample was named CGS.

Preparation of chitosan/cinder-based activated carbon composite membrane (for the synthetic route, see Figure 1): A certain amount of chitosan solution (30.0 mL) was first placed into a three-necked flask, and the system was heated in a 50 °C water bath. Under mechanical stirring, both glutaraldehyde aqueous solution (10.0 mL) and a certain amount of CGS (0.05, 0.10, 0.15, 0.20, 0.30, 0.60, and 1.20 g) were added into the above solution. After stirring for a certain time, a certain amount of the mixture was taken out for the purpose of freeze-drying storage. The serial samples were labeled as F-0.05, F-0.10, F-0.15, F-0.20, F-0.30, F-0.60, and F-1.20, respectively. A blank film without CGS loading was also prepared for comparison and denoted as F-0.

### 2.3. Morphological Characterization

The chemical structure of the films was characterized by the Fourier transform infrared spectroscopy (FT-IR) of the Frontier FT-IR spectrometer (INVENIO, Brook, Germany). The morphological characteristics of different samples were investigated using a field emission scanning electron microscope (FE-SEM, FEI Verios 460). The element compositions and valence states of different samples were examined by X-ray photoelectron spectroscopy (XPS, AXIS SUPRA system). The specific surface area and pore size distribution of the composite membrane were measured by a BET-physical adsorption instrument-specific surface area analyzer (McMeritik (Shanghai) Instrument Co., Ltd., ASAP 2460, Shanghai, China). The freeze-dried composite membrane was added to pure water at 25 °C for standing water adsorption and removed at a certain interval to determine the swelling ratio. The mechanical properties of the composite membrane were tested by a servo electronic tensile tester (high-speed rail, AI-7000-NGD).

### 2.4. RhB and Cr (VI) Adsorption Experiment

The concentrations of Cr (VI) and RhB were tested by UV spectrophotometry using the diphenylcarbazide chromogenic method. In total, 60.0 mL Cr (VI) or RhB solution with a certain concentration were placed in a conical flask. A sample weighing 0.2 g was added to the corresponding conical bottle. Additionally, the conical flask was placed into a constant temperature oscillation box (30 °C, 120 r/min) for oscillation adsorption. After absorption for a certain interval, the suspension of approximately 60.0 mL was removed for centrifugation. After centrifugation, the supernatant was used as the sample to measure the concentrations of Cr (VI) and RhB.
$$\mathrm{Adsorbing}\;\mathrm{capacity}:\;q_{t}=\frac{V\left(C_{0}-C_{t}\right)}{m},$$
$$\mathrm{Removal}\;\mathrm{rate}:\;\eta =\left(1-\frac{C_{t}}{C_{0}}\right)\times 100\%,$$
where *C*_0_ and *C_t_* are the initial mass concentration of RhB and Cr (VI) and the mass concentration at moment *t* (mg/L), respectively; *q_t_* is the adsorption amount of material on RhB and Cr (VI) at moment *t* (mg/g). *V* is the volume of RhB and Cr (VI) added (L), respectively, *m* is the mass of adsorbent material (g), and *η* is the removal rate.

## 3. Results and Discussion

### 3.1. Adsorption Performance Analysis

In order to investigate the adsorption performance of coal gasification slag, 0.2 g of coal gasification slag were added into 60.0 mL (40.0 mg/L) RhB or Cr (VI) solution. It can be seen from Figure 2a that the adsorption of Cr (VI) by coal gasification slag can reach a dynamic equilibrium after approximately 10 min. In this study, the removal rate of Cr (VI) was determined to be 25.0%. As can be seen from Figure 2a, the removal rate of Cr (VI) by blank film F-0 can reach more than 80.5% within 10 min, and the adsorption equilibrium is reached within 40 min. At equilibrium, the removal rate of Cr (VI) was 95.5%, indicating that F-0 has a good adsorption effect on Cr (VI).

It can be seen from Figure 2b that RhB adsorption by coal gasification slag can achieve a 98.0% removal rate within 5 min and a 100% removal rate within 10 min. However, the adsorption effect of blank membrane F-0 on RhB was poor, and the removal rate was less than 5.0% when the adsorption equilibrium was reached (Figure 2b). In conclusion, the coal gasification slag had a good adsorption effect on RhB, but the adsorption effect on Cr (VI) was poor.

The removal results of Cr (VI) by different composite membranes are shown in Figure 3. When the initial concentration of Cr (VI) was 20 mg/L and the adsorption reached the equilibrium state, the adsorbing capacity of Cr (VI) by F-0.30 and F-0 was 5.93 mg/g and 5.85 mg/g, respectively (Figure 3a). When the initial concentration of Cr (VI) was 40 mg/L and the adsorption reached the equilibrium state, the adsorbing capacity of Cr (VI) by F-0.30 and F-0 was 11.78 and 11.46 mg/g, respectively (Figure 3b). When the initial concentration of Cr (VI) was 60 mg/L and the adsorption reached the equilibrium state, the adsorbing capacity of Cr (VI) by F-0.30 and F-0 was 17.66 and 17.51 mg/g, respectively (Figure 3c). When the initial concentration of Cr (VI) was 80 mg/L and the adsorption reached the equilibrium state, the adsorbing capacity of Cr (VI) by F-0.30 and F-0 was 23.45 and 23.16 mg/g, respectively (Figure 3d). When the initial concentration of Cr (VI) was 100 mg/L and the adsorption reached the equilibrium state, the adsorbing capacity of Cr (VI) by F-0.30 and F-0 was 29.31 and 29.13 mg/g, respectively (Figure 3e).

Figure 3f shows that although the RhB adsorbing capacity of the blank film F-0 was less than 0.60 mg/g, the RhB adsorbing capacity of the composite film gradually increased with the amount of coal gasification slag. When the additional amount of coal gasification slag was 0.60 and 1.20 g, the adsorption of RhB by F-0.60 and F-1.20 reached 90% at 60 min and reached the adsorption equilibrium at 120 min, and the adsorbing capacity was 11.20 mg/gat this time. When the adsorption time reached 180 min, the composite membrane with the best adsorption effect was F-0.30, and its adsorbing capacity was 11.44 mg/g. These results indicate that the loading of coal gasification slag can provide chitosan film with the ability to adsorb organic dye RhB.

The above results indicate that loading coal gasification slag can not only provide F-0 with the ability to adsorb organic dye RhB but also affect the adsorption effect of F-0 on Cr (VI), realizing the function of the composite film to adsorb both heavy metal ions and organic dyes simultaneously. The adsorbing capacity of Cr (VI) by F-0.30 at 2 min was significantly higher than that of the F-0 (Table 1), indicating that the introduction of CGS can accelerate the adsorption of Cr (VI) by the composite membrane. This conclusion was also reflected in the equilibrium time of different concentrations of Cr (VI) adsorbed by different samples.

### 3.2. Adsorption of Cr (VI) in Mixed Metal Ions

The adsorption results of the composite membrane on mixed metal ions are shown in Figure 4. The concentrations of Cd (II) and Cu (II) in the mixed solution did not decrease after adding the adsorbent, indicating that the adsorption effect of the composite membrane on Cd (II) and Cu (II) was not obvious (Figure 4a,b). The initial concentration of Pb (II) in the mixed solution was lower (Figure 4d) due to the reaction between Pb (II) and SO_4_^2−^ in the mixed solution to form PbSO_4_ precipitate, resulting decreased Pb (II) concentration. In the mixed solution, the adsorption effect of the composite membrane on Cr (VI) was significantly better than that of other metal ions, and the concentration of Cr (VI) in the mixed solution decreased from 1.6 mg/L to almost zero, reflecting the selective adsorption of the composite membrane on Cr (VI) (Figure 4c).

### 3.3. Other Influencing Factors

The effects of temperature, pH, and adsorbent dosage on Cr (VI) adsorption were investigated (Figure 5). The adsorption effect of F-0 on Cr (VI) was similar to that of F-0.30 when the pH was neutral (6~7), and the adsorption amount of F-0.30 on Cr (VI) remained stable when the pH fluctuated (acidic or alkaline), close to 12.0 mg/g. This was due to the presence of CGS, which helped F-0.30 buffer the external pH changes (Figure 5a). As the ambient temperature increased, the adsorption of Cr (VI) by F-0 and F-0.30 showed a decreasing trend (Figure 5b), indicating that the adsorption of Cr (VI) by F-0 and F-0.30 is an exothermic reaction. The adsorption capacity of the adsorbent decreased with the increase in the amount of adsorbent added, but the removal rate increased (Figure 5c). This is because when the content of pollutants in the solution is fixed, the amount of adsorbent added increases, and the amount of adsorption per unit mass decreases. The different initial concentrations of RhB also affect the adsorption of RhB on the composite membrane (Figure 5d). As the initial concentration of RhB increased, the overall trend of the RhB removal rate of F-0 and F-0.30 gradually decreased. This is because when the initial concentration of RhB is low, the composite membrane does not reach the adsorption saturation state, which can adsorb RhB well. When the initial concentration of RhB further increases, the adsorption site on the surface is full. RhB will further transfer to the interior of the material. However, due to the blockage of the surface, RhB cannot transfer further to the interior of F-0.30, so the adsorption amount decreases [22]. In this work, the main role of RhB adsorption was in the coal gasification slag, and relevant research shows that the material for RhB was in a neutral pH range (6~8), so this article does not discuss it in detail [23].

### 3.4. Analysis of the Relationship between Structure and Adsorption Property

Figure 6a shows the infrared spectra of chitosan (Cs), blank film F-0, coal gasification slag CGS, and F-0.30. In the spectrum of sample CGS, fewer groups were found because the CGS needs to be calcined at 800 °C in the preparation process, and the peak at 1090 cm^−1^ belonged to the shear vibration peak of the CH-CH_2_ on the Si-CH-CH_2_ group. The wide peak at 3120 cm^−1^ was related to the stretching vibration peak of C-H. The F-0 showed a stretching vibration peak of C=N at 1538 and 1635 cm^−1^ [18], indicating that the Schiff base is formed by the reaction between chitosan and glutaraldehyde (Figure 6b) [24]. The stretching vibration peak of C-O-C was also found at 1040 cm^−1^. The stretching vibration peak of primary amine C-NH_2_ was located at 1395 cm^−1^. CGS loading had no obvious effect on the infrared peak of the composite film, which indicates that CGS has no destructive effect on the structure of the composite film. It also indicates that the binding modes of CGS and composite film are mainly physical embedding and hydrogen bond cross-linking. This conclusion was verified in the following characterization analysis.

Figure 7 is the SEM image of the prepared sample. Figure 7a shows that the surface of F-0 is flat and smooth with a small number of stacked ridges, which is in line with the typical characteristics of freeze-dried samples. Figure 7b shows that the interior of F-0 exhibits an irregular porous structure with pore sizes ranging from 20 to 80 µm. This is due to the fact that F-0 can sublimate the solid water in the material directly to gaseous water during the freeze-drying process. This allows the material to retain the shape it had when it was frozen while leaving empty spaces where the original solid ice would have been, and these conditions promote the formation of a laminar porous structure in F-0. Figure 7c clearly shows that the surface of F-0.30 is inlaid with CGS spherical particles, indicating that CGS is successfully loaded in the structure of the composite film. Figure 7d shows that the interior of F-0.30 presents a three-dimensional network structure. It can be clearly seen that the AC is attached to the inner layer of the composite film, which further proves that CGS is successfully loaded in the inner layer of the composite film. Compared with the internal structure of F-0, the pore size of the internal structure of F-0.30 is larger and reaches up to 110 µm, indicating that the load of CGS can stretch the pore structure of the composite membrane, which is beneficial for pollutants to enter the membrane and maintain full contact with the membrane material, thus accelerating the removal of pollutants.

According to the results of SEM, we can speculate that CGS loading may increase the specific surface area of the composite film, and the adsorption performance of the membrane material is closely related to its specific surface area. As can be seen from Figure 8a, the specific surface area of the composite film is significantly increased after adding CGS, and the specific surface area of F-0 is calculated to be 0.4139 m^2^/g, but that of F-0.30 is 6.6996 m^2^/g. Figure 8b shows that the number of F-0 nanopores is relatively small, and the number of F-0.30 nanopores is significantly increased after adding CGS, indicating that adding CGS can stretch the nanopores of the composite film, which is conducive to improving the adsorption capacity and adsorption rate of pollutants of the composite film. This conclusion can be further illustrated by the swelling property. Figure 8c shows the swelling properties of different samples. It can be seen from Figure 8c that the equilibrium swelling rate of the composite film decreases to a certain extent after loading CGS, indicating that the interaction force between CGS and the composite film is hydrogen bonds, which is consistent with the infrared spectrum analysis results above (Figure 6b). CGS occupied a certain number of hydrogen bond-forming sites in the composite film, resulting in a decrease in the swelling rate of the composite film. However, the time for the composite membrane to reach swelling equilibrium was shorter after loading CGS. That is to say, the adsorption rate of the composite membrane to the aqueous solution was improved. The reason is that the CGS load increased the pore size of the membrane inside the membrane, thus accelerating the water absorption rate of the composite membrane. This was also the reason why the adsorption rate of Cr (VI) of the composite membrane was faster than that of the F-0.

### 3.5. Adsorption Kinetics and Mechanism Analysis

Figure 9 shows the fitting diagram of the adsorption kinetics of Cr (VI) on the F-0 and the composite membrane. Figure 10 shows the finishing of the fitting results. It can be seen from Figure 9a and Figure 10c that the adsorption of Cr (VI) by F-0 and the composite membrane does not conform to the first-order kinetics because the correlation coefficient R^2^ between the fitting results and the first-order kinetics is less than 0.9. The correlation coefficients R^2^ between the fitting results and the second-order kinetics are all greater than 0.9, indicating that the adsorption process of Cr (VI) by F-0 and the composite membrane conforms to the second-order kinetics and the equilibrium adsorption capacity (q_e,2_) calculated based on the fitting results of the second-order kinetics is closer to the actual measured adsorption capacity (q_e,exp_) (Figure 10a,b). Additionally, the value of k for second-order dynamics is greater than that for first-order dynamics (Figure 10d). Generally speaking, second-order kinetics involves the sharing or transfer of electron pairs between the adsorbent and adsorbent, so the adsorption process of Cr (VI) by blank and composite membranes belongs to the chemisorption and may involve the formation of coordination bonds. To verify this conclusion, we tested the X-ray photoelectron spectroscopy (XPS) of the F-0 samples before and after Cr (VI) adsorption.

Figure 9c shows the full spectrum of XPS before and after F-0 adsorbed Cr (VI). It can be seen from the figure that there are characteristic peaks of C, N, and O before and after F-0 adsorbed Cr (VI), and the characteristic peak of Cr_2p_ appears after adsorption. Figure 9d shows the high-resolution Cr_2p_-XPS map. The XPS peaks at 579.98 eV and 583.94 eV binding energy correspond to Cr (VI) in Cr_2_O_7_^2−^ and CrO_4_^2−^ [25]. However, we found that the binding energy positions of peaks were significantly larger than those of the standard binding energy, indicating that the electrons adsorbed by Cr_2_O_7_^2−^ may have been transferred to F-0. Figure 9e shows the high-resolution C_1s_-XPS map. The XPS peak at 285.78 eV corresponds to C-OH in chitosan. After the adsorption of Cr (VI), the binding energy of the corresponding peak of C-OH increases, indicating that the electron cloud around C-OH decreases, and the O-supplied electron pair in the reaction C-OH forms a coordination bond with Cr (VI). Figure 9f shows the high-resolution N_1s_-XPS map. The XPS peaks at 400.51 eV and 399.55 eV correspond to imino (-NH for short) and nitrogen-to-carbon single bonds (N-C) in F-0, respectively. After the adsorption of Cr (VI), the binding energy of the corresponding peak of -NH increases, indicating that the electron cloud around -NH decreases, and the N-supplied electron pair in -NH forms a coordination bond with Cr (VI). Meanwhile, the binding energy of the corresponding peak of N-C is decreased and shifted, indicating that the electron cloud density on the N-C structure is increased. The reason is that the electrons of Cr_2_O_7_^2−^ can be transferred to F-0 after the formation of chemical bonds between Cr (VI) and C-OH and -NH, which is consistent with the previous high-resolution Cr_2p_-XPS map analysis results. The above XPS analysis results are in good agreement with the kinetic analysis results.

Figure 11 shows the fitting diagram of the adsorption kinetics of RhB on blank film F-0 and composite film. Figure 12 shows the fitting results. Because F-0 does not adsorb RhB, we did not analyze its adsorption kinetics. It can be seen from Figure 11a,b and Figure 12c that the adsorption of RhB by composite membrane conforms to first-order and second-order kinetics because the correlation coefficient R^2^ between the fitting results and first-order and second-order kinetics is close to 0.9. Moreover, the equilibrium adsorption capacity (q_e_) calculated from the fitting results of first-order kinetics and second-order kinetics is very close to the actual measured adsorption capacity (q_e,exp_) (Figure 12a,b). Interestingly, the second-order kinetic constant k is greater than the first-order kinetic constant when the loading of CGS is low, and the value of first-order kinetic k is greater than the second-order kinetic when the loading of CGS is high (Figure 12d), which indicates that the physical adsorption capacity of the composite membrane is stronger when the loading of CGS is increased. According to the above results, we speculate that the adsorption of RhB on CGS is between physical adsorption and chemical adsorption, and the adsorption mode is mainly hydrogen bonds. Because the blank film F-0 does not adsorb RhB, the adsorption of RhB by the composite film may be due to the presence of coal gasification slag (CGS). Therefore, in order to further analyze the mechanism of RhB adsorption by the composite membrane, we analyzed the Zeta potential, Raman spectrum, and infrared spectrum before and after CGS adsorption of RhB.

The Zeta potential value of CGS is −1.41 mV (Figure 11c), and the whole material is electronegative. Because RhB is a cationic dye, if it is absorbed in the form of ions, the Zeta potential value of CGS will increase after adsorbing RhB. However, the electronegativity of CGS increases to −1.38 mV after the adsorption of RhB, which is almost unchanged compared with that before adsorption. This indicates that RhB may be adsorbed on the surface of CGS in molecular form. Figure 11d shows the Raman spectra before and after CGS adsorption of RhB. As can be seen from the figure, the spectral bands at 1315 cm^−1^ and 1600 cm^−1^ are assigned to the D peak and G peak of CGS, respectively. The D peak represents the edge and defect site of the molecule (or lattice) containing carbon, and the G peak represents the in-plane stretching vibration of C hybridized by carbon atom sp2 [26]. CGS and RhB are carbon-containing substances, and both have sp2 hybrid C. After the adsorption of RhB, the intensity of peak D and peak G of CGS increases, indicating that RhB is adsorbed on the surface or inside of CGS, leading to increasing the carbon content in CGS. However, these two peaks do not undergo chemical shifts, indicating that the adsorption of RhB in the CGS structure is not chemisorption either. Figure 11e shows the FT-IR spectra before and after RhB adsorption by CGS. Before the adsorption of RhB, the infrared peaks at 3120 cm^−1^, 1390 cm^−1^, and 1636 cm^−1^ are the stretching vibration peaks of the C-H bond in CGS, the in-plane bending vibration peak of the C-H bond, and the stretching vibration peak of C = C bond on olefin. After the adsorption of RhB by CGS, these peaks of CGS do not change significantly, and two small infrared absorption peaks appeared at 3441 cm^−1^ and 1134 cm^−1^, both of which are -OH on RhB [27]. Other functional groups of RhB do not appear on FT-IR spectra, indicating that RhB is mainly absorbed in the pores of CGS. The reason for this phenomenon is that the pore structure of CGS is very small (2–5 nm), with a capillary phenomenon.

### 3.6. Mechanical Strength and Cyclic Performance Investigation

From Figure 13a, it can be obtained that F-0 can withstand fracture stress of 388.41 N/mm, and F-0.30 can withstand fracture stress of 851.42 N/mm. It can be seen that the tensile capacity of the composite film is enhanced by 2.2 times after adding CGS, and it can be seen that CGS can improve the mechanical properties of the composite film to a certain extent. From the adsorption cycle of the composite membrane, Figure 13b shows that the adsorption rate and removal rate decrease after six cycles of F-0. After loading CGS, the material has a better adsorption cycle effect, and the removal rate of Cr (VI) by F-0.30 can still reach approximately 90% after six cycles of adsorption resolution. Figure 13c is the RhB adsorption cycle graph. It can be seen that after six cycles of adsorption, the removal rate of both RhB by F-0 and F-0.30 decrease to some extent, but the removal rate of F-0.30 is still 46.8% in the sixth cycle, while that of F-0 is only approximately 10%, and both of them have a better effect on the adsorption of RhB compared with that of F-0.3. The above results prove that the introduction of coal gasification slag can enhance the recycling performance of the composite membrane.

## 4. Review of Adsorption Mechanism and Comparison of Properties

A schematic diagram of the adsorption mechanism of the composite membrane for RhB and Cr (VI) is shown in Figure 14, from which it can be seen that the experimentally prepared CGS was loaded on the chitosan composite membrane, and it can be proved by the data of BET, tensile force, and SEM that the composite membrane loaded with CGS has the advantages of increased specific surface area, enhanced mechanical properties, and increased nanopores, which provide more adsorption sites for the adsorption of pollutants. After the adsorption of Cr (VI), the binding energy of the corresponding peak of C-OH increases (Figure 9e), indicating that the electron cloud around C-OH decreases and the O in C-OH provides electron pairs to form coordination bonds with Cr (VI), while the N spectrum shows that the N in -NH provides electron pairs to form coordination bonds with Cr (VI) (Figure 9f), and Cr (VI) is formed as Cr_2_O_7_^2−^ and CrO_4_^2−^ etc. RhB is adsorbed in the molecular form on the surface and in the void of CGS, and if it is adsorbed in ionic form in CGS, the electronegativity of CGS increases after adsorption of RhB, while the electronegativity of CGS changes very little before and after adsorption. Then, based on the physical and chemical characteristics of CGS, the abundant pore size and the large specific surface area, it was inferred that the adsorption of RhB by CGS was physical adsorption. This is consistent with the simulation calculation results of adsorption kinetics. Therefore, when pollutants such as RhB and Cr (VI) in water come into contact with the composite membrane, the amino group on the molecular chain of chitosan can adsorb Cr (VI) by coordination reaction, while the hydrogen bonding and void space of CGS can adsorb RhB in water. The composite membrane can adsorb two different pollutants by hydrogen bonding and chemical interaction and thus can remove the pollutants in water.

The following is the comparison of adsorption properties between other similar materials and materials prepared in this work (Table 2).

## 5. Conclusions

In this study, a composite membrane was successfully prepared by a one-step method using chitosan and coal gasification slag as raw materials. From SEM images, the prepared composite membrane has a three-dimensional network structure, and coal gasification slags are stably loaded in the network structure of the composite membrane. The Cr (VI) adsorption rate of the composite membrane can reach 98.9%, and the Cr (VI) adsorption rate can be accelerated by coal gasification slag. During RhB adsorption, the adsorption rate of the blank membrane is only 5%, while the RhB adsorption rate of the composite membrane can reach 95.3%. After the analysis of the specific surface area and pore size distribution, it was found that the specific surface area of the composite membrane was 16.2 times higher than that of the blank membrane. Meanwhile, the coal gasification slag could spread the nanopores of the composite membrane; thus, coal gasification slag could improve the adsorption capacity and adsorption rate of the composite membrane for pollutants. After the analysis of adsorption kinetics and adsorption mechanism, it was found that Cr (VI) is adsorbed mainly by forming the coordination bond with the amino group on the chitosan molecular chain, while RhB is adsorbed by forming the hydrogen bond with gasification residue surface. Moreover, coal gasification slag can increase the mechanical properties of the composite film by approximately three times, and the adsorption removal rate of Cr (VI) is close to 90% after six cycles of application to improve the practical application value of composite membrane.

## Figures and Tables

**Figure 1 molecules-27-07173-f001:**
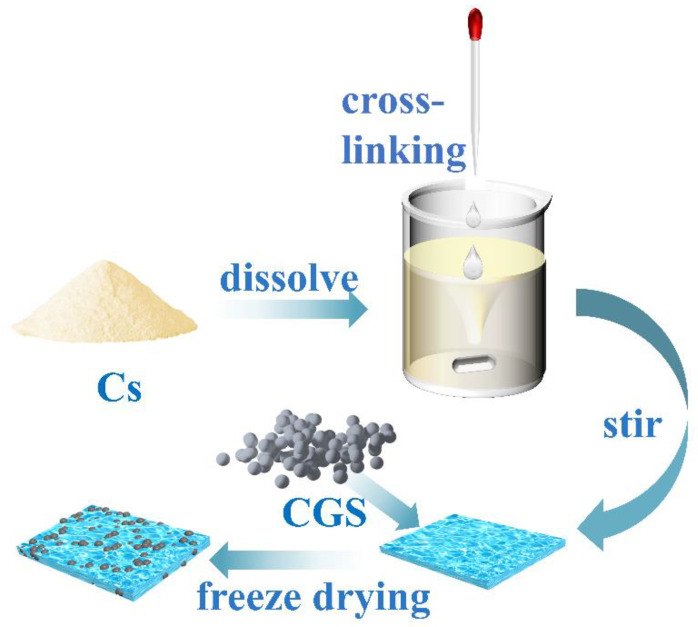
Route for the chitosan/CGS composite membrane.

**Figure 2 molecules-27-07173-f002:**
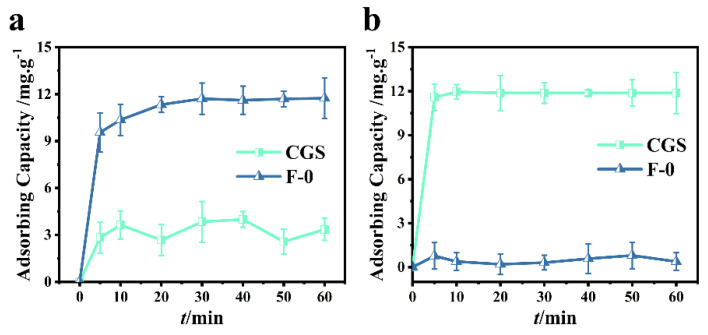
(**a**) Cr (VI) adsorption over CGS and F-0. Reaction conditions: Cr (VI) (40 mg/L, 60 mL), adsorbent (0.2 g). (**b**) RhB adsorption over CGS and F-0. Reaction conditions: RhB (40 mg/L, 60 mL), adsorbent (0.2 g).

**Figure 3 molecules-27-07173-f003:**
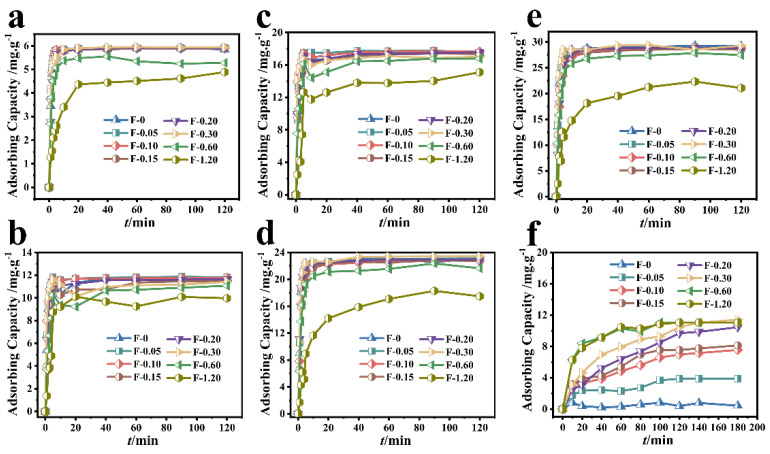
Different composite membranes adsorbed different initial concentrations of Cr (VI). (**a**–**e**) the initial concentrations of 20, 40, 60, 80, and 100 mg/L, respectively. Reaction conditions: Cr (VI) (60 mL), adsorbent (0.2 g). (**f**) The adsorption results of different composite membranes on RhB. Reaction conditions: RhB (40 mg/L, 60 mL), adsorbent (0.2 g).

**Figure 4 molecules-27-07173-f004:**
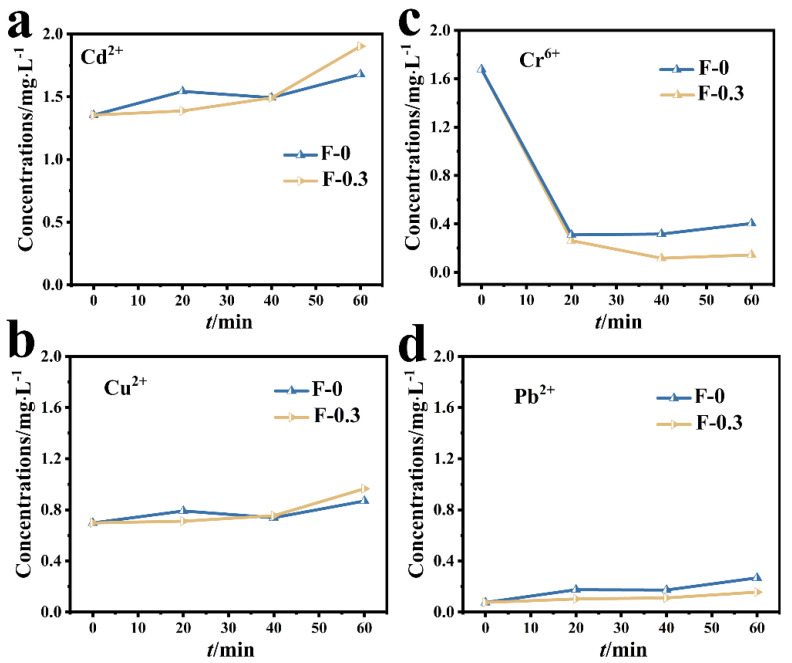
(**a**) Changes in the concentration of Cd (II) in the mixed solution; (**b**) changes in the concentration of Cu (II) in the mixed solution; (**c**) changes in the concentration of Cr (VI) in the mixed solution; (**d**) changes in the concentration of Pb (II) in the mixed solution.

**Figure 5 molecules-27-07173-f005:**
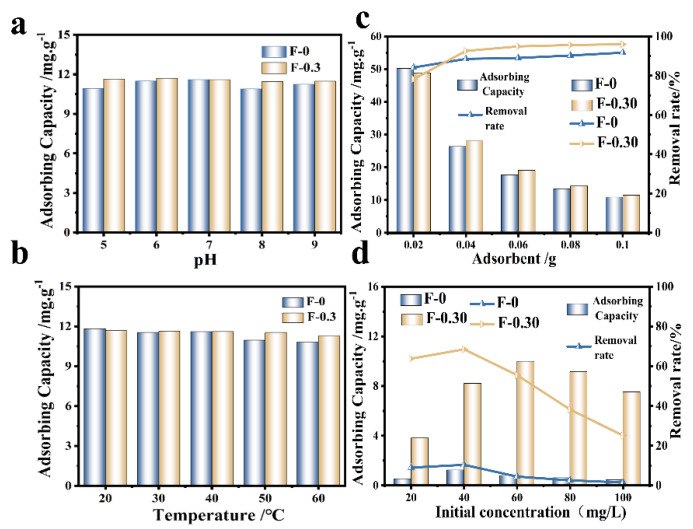
(**a**) The variation of Cr (VI) adsorption by an adsorbent at different pH. Reaction conditions: Cr (VI) (40 mg/L, 30 mL), adsorbent (0.1 g), and temperature (20 °C). (**b**) The variation of Cr (VI) adsorption by adsorbent at different temperatures. Reaction conditions: Cr (VI) (40 mg/L, 30 mL), adsorbent (0.1 g), and pH (No adjustment). (**c**) The adsorption results of Cr (VI) by adding adsorbent. Reaction conditions: Cr (VI) (40 mg/L, 30 mL), pH (not adjustable), and temperature (20 °C). (**d**) The effect of initial concentration on the removal rate and adsorption capacity of RhB. Reaction conditions: 30.0 mL. Reaction time: 60 min.

**Figure 6 molecules-27-07173-f006:**
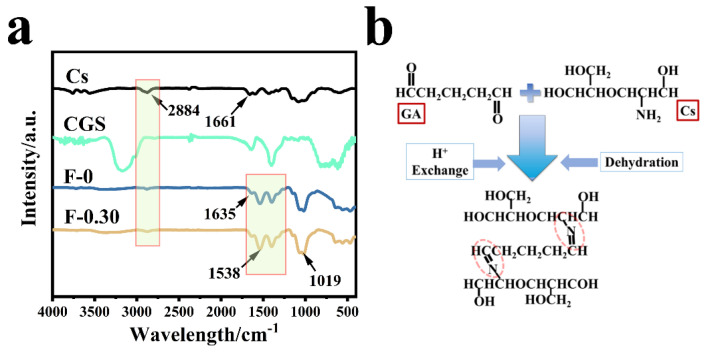
(**a**) The infrared spectra of different samples; (**b**) the formation process of Schiff base.

**Figure 7 molecules-27-07173-f007:**
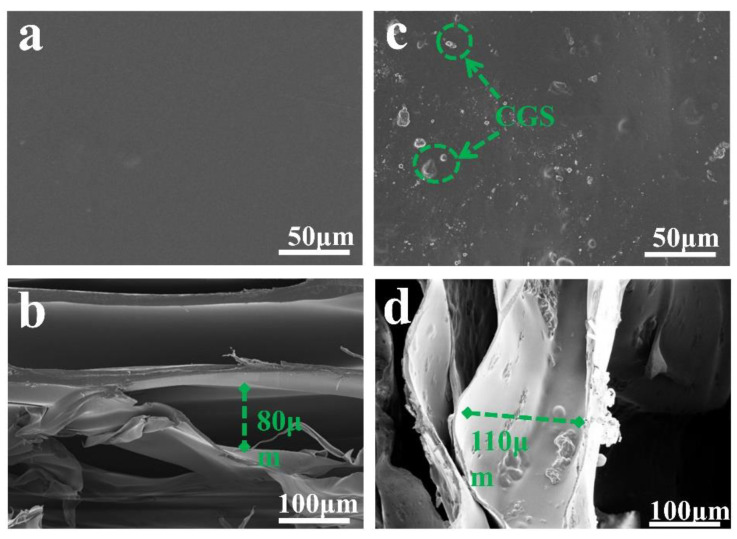
(**a**) The surface of F-0, (**b**) the interior of F-0, (**c**) the surface of F-0.3, and (**d**) the interior of F-0.3.

**Figure 8 molecules-27-07173-f008:**
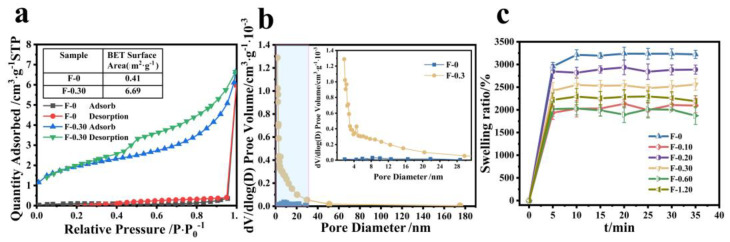
(**a**) The specific surface area of F-0 and F-0.3, (**b**) the pore size distribution of F-0 and F-0.3, and (**c**) the swelling rate of the samples. Reaction conditions: distilled water (30 mL), composite membrane (0.1 g).

**Figure 9 molecules-27-07173-f009:**
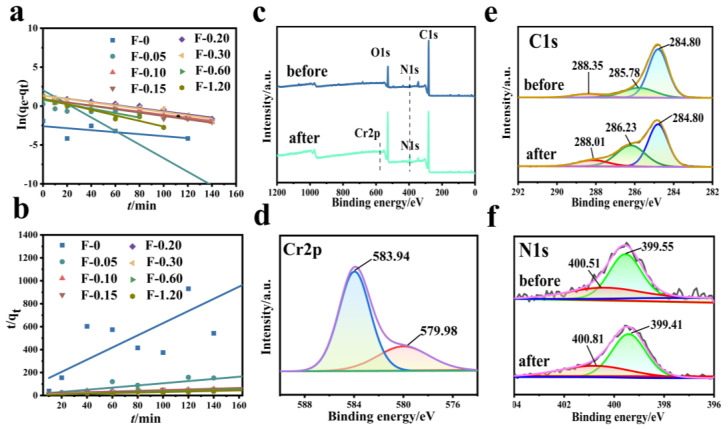
(**a**) First-order kinetic simulation of Cr (VI) adsorption by the composite membrane. Reaction conditions: Cr (VI) (20 mg/L, 60 mL), composite membrane (0.2 g). (**b**) Second-order kinetic simulation of Cr (VI) adsorption by the composite membrane. Reaction conditions: Cr (VI) (20 mg/L, 60 mL), composite membrane (0.2 g). (**c**–**f**) XPS spectra of F-0 before and after the Cr (VI) adsorption.

**Figure 10 molecules-27-07173-f010:**
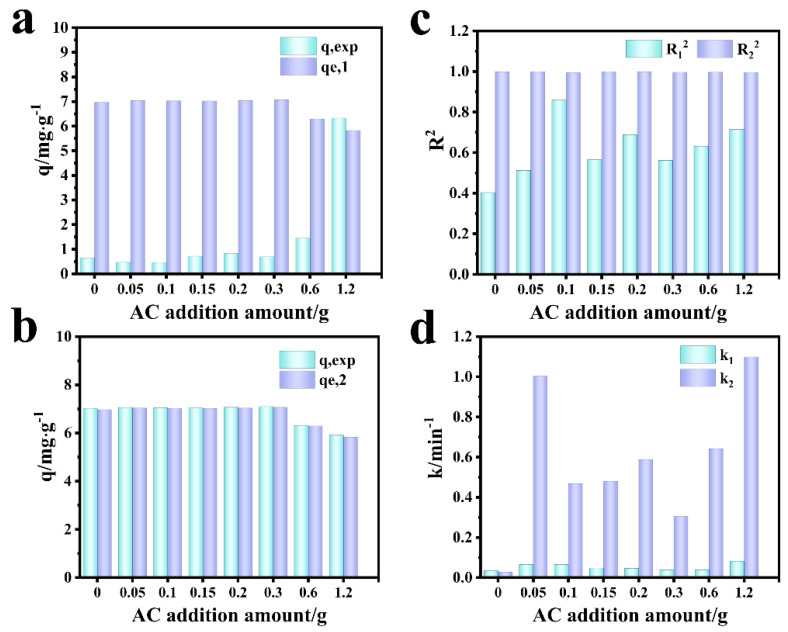
Comparison of simulation results of Cr (VI) adsorption kinetics by composite membrane. (**a**) First-order dynamics q_e,1_ vs. q_exp_. (**b**) Second-order dynamics q_e,2_ vs. q_exp_. (**c**) R_1_^2^ vs. R_2_^2^; (**d**) k_1_ vs. k_2_.

**Figure 11 molecules-27-07173-f011:**
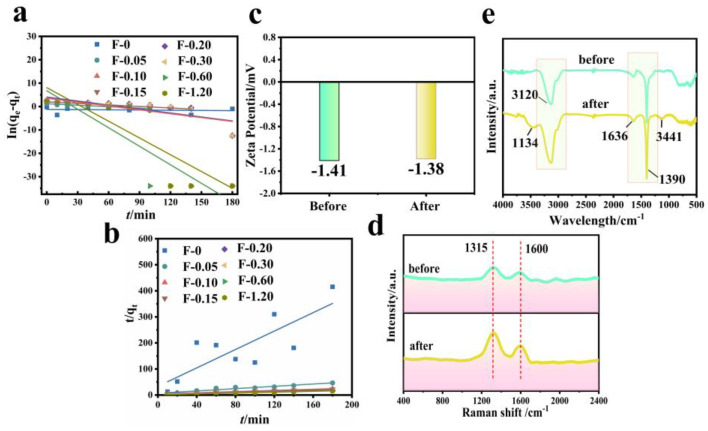
(**a**) First-order kinetic simulation of RhB adsorption by composite membrane. Reaction conditions: RhB (40 mg/L, 60 mL), composite membrane (0.2 g). (**b**) Second-order kinetic simulation of RhB adsorption by composite membrane. Reaction conditions: RhB (40 mg/L, 60 mL), composite membrane (0.2 g). (**c**) Zeta potential before and after RhB adsorption by CGS. Reaction conditions: RhB (40 mg/L, 60 mL), CGS (0.2 g). (**d**) Raman spectrum before and after RhB adsorption by CGS. Reaction conditions: RhB (40 mg/L, 60 mL), CGS (0.2 g). (**e**) FTIR spectrum before and after RhB adsorption by CGS. Reaction conditions: RhB (40 mg/L, 60 mL), CGS (0.2 g).

**Figure 12 molecules-27-07173-f012:**
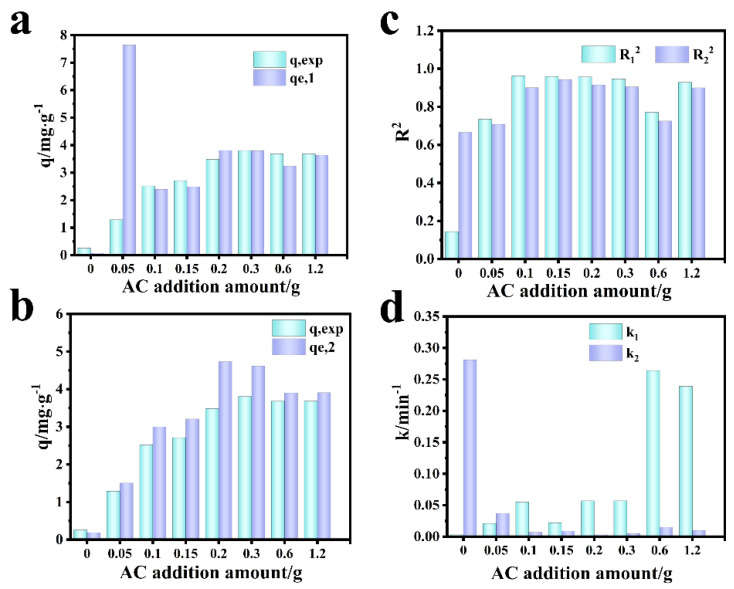
Comparison of simulation results of adsorption kinetics of RhB by the composite membrane. (**a**) First-order dynamics q_e,1_ vs. q_exp_. (**b**) Second-order dynamics q_e,2_ vs. q_exp_. (**c**) R_1_^2^ vs. R_2_^2^. (**d**) k_1_ vs. k_2_.

**Figure 13 molecules-27-07173-f013:**
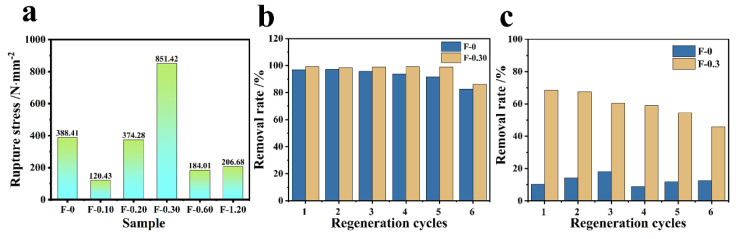
(**a**) The tensile ability test of the composite membrane. (**b**) The Cr (VI) adsorption cycle results of F-0 and F-0.3. Reaction conditions: Cr (VI) (40 mg/L, 60 mL), adsorbent (0.2 g), and reaction time (60 min). (**c**) The RhB adsorption cycle results of F-0 and F-0.3. Reaction conditions: RhB (40 mg/L, 60 mL), adsorbent (0.2 g), and reaction time (60 min).

**Figure 14 molecules-27-07173-f014:**
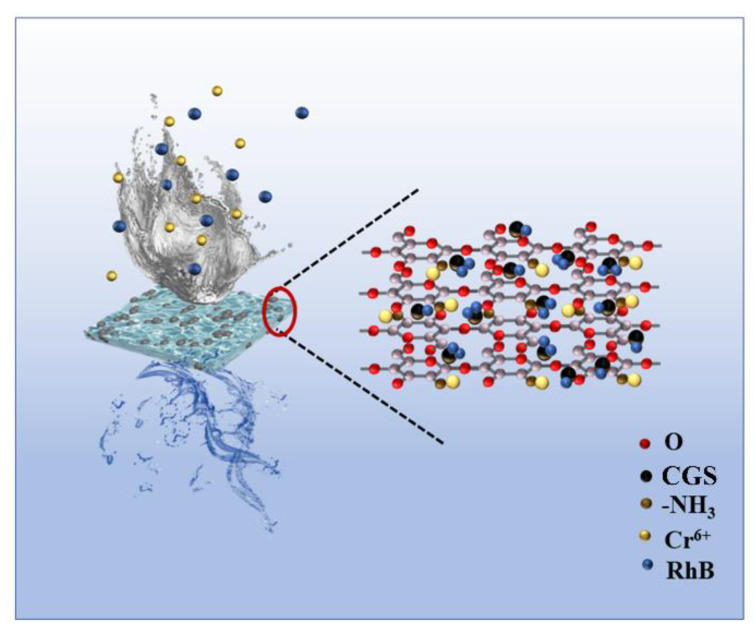
Schematic diagram of adsorption mechanism of RhB and Cr (VI) by the composite membrane.

**Table 1 molecules-27-07173-t001:** Comparison of Cr (VI) adsorbing capacity of F-0 and F-0.3 at 2 min and equilibrium.

Sample	Initial Concentration (mg/L)	Adsorbing Capacity (mg/g)	
		2 min	End
F-0.3	20	5.12	5.85
	40	10.06	11.78
	60	15.41	17.66
	80	19.32	23.45
	100	24.03	29.31
F-0	20	4.87	5.85
	40	9.05	11.46
	60	13.82	17.51
	80	18.22	23.16
	100	21.96	29.13

**Table 2 molecules-27-07173-t002:** Comparison of adsorption of pollutants by different composite films.

Author	Membrane Materials	Contaminants	Removal Rate (%)	Adsorption Amount (mg/g)	Literature
Gharbani	CS, graphite carbon nitride/polyvinylidene difluoride	RhB	72.7 (2 mg/L, pH = 3, and 2.0 g g-C_3_N_4_)	4.16	[28]
Zhao	PVDF, CS, CNTs-COOH	RhB	99.0 (10 mg/L, pH = 2, dosage 0.5 g)	2.0	[29]
Kirisenage	acrylate polymer nanostructured graphitic carbon	As		1.5	[30]
		NH4+		0.27	
Park	chitosan-coated iron oxide nanoparticles	Cr (VI)		14.45 (intermittent system)	[31]
				14.1 (continuous inflow system)	
Queirós	Al(OH)_3_/PVDF-HFP MIL-88-B(Fe)/PVDF-HFP UiO-66-NH_2_/PVDF-HPP	Cr (VI)	12	5.0	[32]
			62	3.0	
			97 (5 mg/L)	3.0	
This article	CS/CGS	Cr (VI)	97.7 (100 mg/L, 60 mL, Adsorbent (0.2 g))	50.0	This article
		RhB	96.2 (40 mg/L, 60 mL, 0.2g)	11.5 (40 mg/L, 60 mL, 0.2 g)	This article

## Data Availability

Not available.

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
