# Peer review of "Preparation of a Chitosan/Coal Gasification Slag Composite Membrane and Its Adsorption and Removal of Cr (VI) and RhB in Water"

_molecules, 2022, doi:10.3390/molecules27217173_

Round 1
Reviewer 1 Report
The work reported in this manuscript is interesting and well presented. However, it requires corrections and improvements before the acceptance. The work requires revision. Some comments are:
1. The abstract is concise and accurately summarizes the essential information. Abstract should be rewritten to summarize the work; the abstract should briefly state the purpose of the research, the principal results, and major conclusions. An abstract is often presented separately from the article, so it must be able to stand alone.
2. Units in all sections should be uniform, Significant figures should be uniform,
3. References must be cited in the correct place in the text, and also must match correctly their position in the list. Please Cite references at appropriate locations and list them correctly. Spell of references must be checked.
4. Ensure that all figures are cited in the text.
5. More elaboration on the chemical interaction between the components is required.
6. The introduction; what is the gab to cover
7. Morphology images please deeply discuss the images with respect to structure
8. Captions of the figure and tables must be with complete information, conditions etc
9. Please improve the conclusion with clear quantitative findings
Reviewer 2 Report
Dear Editor
The manuscript presents, Preparation of Chitosan/Coal Slag Based Activated Carbon Composite Membrane and Its Adsorption and Removal of Cr (VI) and RhB in Water, which is interesting. The subject addressed is within the scope of the journal. The paper is well written and the scope of the work is highly appealing. The experimental part is presented accurately and in sufficient detail. The results are presented precisely and with high scientific competence. Although, the manuscript lacks the novelty and originality that is necessary for a publication; additionally the manuscript does not contain enough deep discussion with already published works. Also, in the conclusions, in addition to summarising the actions taken and results, please strengthen the explanation of their significance. It is recommended to use quantitative reasoning comparing with appropriate benchmarks, especially those stemming from previous work. However, the manuscript, in its present form, contains several weaknesses. Appropriate revisions to the following points should be undertaken in order to justify recommendation for publication.
Considering the review, I suggest that the paper can be accepted with major revision. Herein my main observations:
1. Keywords: Suggesting do not use keywords that are in the title, do not use abbreviation in the keywords,
2. More quantitative information should be provided in the abstract.
3. It is better to rewrite the introduction; first, the problem is described and then solutions are given to reach the nanoparticles. In fact, emphasize the importance of the subject of your article.Review resources and innovate your work towards them conclude by introducing your research objectives. It should be clarified which is the innovative factor of the article. What are the differences of your work with other similar reported works? In the introduction, it needs to connect the state of the art to the paper goals. Please follow the literature review by a clear and concise state of the art analysis. This should clearly show the knowledge gaps identified and link them to the paper goals.
4. The concussion should be concise and to the points indication the application of the work.
5. What materials have been used? Vendor information must be given consistently and completely.
6. What about effect of other parameters such as temperature, pH, initial concentration of Cr and RhB…. on the adsorption.
7. Figures 3,6,8,10 are not clear, replace them.
8. Competition of other heavy metal ions and other dye such as anionic dye should be considered since most of the wastewater contained not only a kind of heavy metal. Authors should also give a comparison of Cr and RhB adsorption capacity between the synthesized sorbent and other reported adsorbents.
9. Rechecked the figures in the caption and in the text
10. The discussion section in the present form is relatively weak and should be strengthened with more details and justifications. Moreover, the manuscript could be substantially improved by relying and citing more on recent literatures,
11. Where is the tables 4 and 5 (line 273), also 2 and 3…. Check the manuscript carefully
12. Is it possible to add isotherm and thermodynamic models?
13. the mechanism need further elucidated
14. It is also better to compare with those related works reported recently, especially, 2D related materials.
15. The conclusion section must be improved. In the conclusions, in addition to summarising the actions taken and results, please strengthen the explanation of their significance. It is recommended to use quantitative reasoning comparing with appropriate benchmarks, especially those stemming from previous work. Pay attention don’t repeat the results in the conclusion.
Round 2
Reviewer 2 Report
Since the respected author has made most of the comments, the article is acceptable